

# Long-term observation of mid-latitude quasi 2-day waves by a water vapor radiometer

Martin Lainer[1], Klemens Hocke[1,2], and Niklaus Kämpfer[1,2]

[1]Institute of Applied Physics, University of Bern, Bern, Switzerland
[2]Oeschger Center for Climate Change Research, University of Bern, Bern, Switzerland

*Correspondence to:* Martin Lainer (martin.lainer@iap.unibe.ch)

**Abstract.** A mesospheric water vapor data set obtained by the middle atmospheric water vapor radiometer (MIAWARA) close to Bern, Switzerland ($46.88°$ N, $7.46°$ E) during October 2010 to September 2017 is investigated to study the long-term evolution and variability of quasi 2-day waves (Q2DWs). We present a climatological overview and an insight on the dynamical behavior of these waves with the occurring spectrum of periods as seen from a mid-latitude observation site. Such a large and nearly continuous measurement data set as ours is rare and of high scientific value. The core results of our investigation include that the activity of the Q2DW manifests in burst-like events and is higher during winter months (November–February) than during summer months (May–August) for the altitude region of the mesosphere (up to $0.02$ hPa in winter and up to $0.05$ hPa in summer) that is accessible for the instrument. Single Q2DW events reach at most about $0.8$ ppm in the $H_2O$ amplitudes. Further, monthly mean Q2DW amplitude spectra are presented and reveal a high frequency variability between different months. A large fraction of identified Q2DW events ($20\%$) develop periods between 38–40 h. Further, we show the temporal evolution of monthly mean Q2DW oscillations continuously for all months and separated for single months over 7 years. The analysis of autobicoherence spectra gives evidence that the Q2DW occasionally is to a high degree phase coupled to diurnal oscillations and to waves with a period close to $18$ h.

## 1 Introduction

The middle atmosphere is the part of Earth's atmosphere that extends from about 10 to 110 km altitude. The upper part (60–110 km) is referred to as MLT (Mesosphere Lower Thermosphere) which is dominated by the interplay of atmospheric waves, including tides, gravity and planetary waves. Important source regions for atmospheric waves seen in the MLT are often found lower in the atmosphere. With decreasing pressure, respectively air density, upward propagating waves are forced to increase their amplitudes (Andrews et al., 1987). This increase in amplitudes can lead to wave breaking and the deposition of momentum which in turn supplies the driving force for large scale residual circulations, like the Brewer-Dobson circulation (Brewer, 1949; Dobson, 1956). Besides the diurnal and semi-diurnal waves, the quasi 2-day wave (Q2DW) is among the strongest wave phenomena within the middle atmosphere. Quasi 2-day waves originate primarily from baroclinic instabilities which can be found in the vicinity of jet streams such as the summertime mesospheric easterly jet. Many studies indicate that these atmospheric regions produce fast-emerging instabilities coupling to the zonal wave number 3 global Rossby-gravity mode





(Salby, 1981; Lieberman, 1999; Rojas and Norton, 2007). Q2DW structures in middle atmospheric temperature observations were first discovered by Rodgers and Prata (1981). Before that time quasi 2-day oscillations were only found in wind data at meteor heights (Muller and Nelson, 1978; Salby and Roper, 1980). Q2DWs not only manifest in wind or temperature fields. Teitelbaum et al. (1981) analyzed one of the first observations of 2-day planetary-wave signatures in atmospheric airglow. A

recent numerical GCM (general circulation model) investigation by Egito et al. (2017) brought new insights on the planetary-wave-induced airglow variability in the mesosphere and lower thermosphere. In regard of the 2-day variability prominent oscillations were found in this simulation during summer at a northern hemispheric mid-latitude ($43° \, N$, $143° \, E$). Usually the Q2DW gets amplified in temporal proximity to the solstices (Wu et al., 1996). For the Northern Hemisphere (NH) the months July and August (after summer solstice) are favored to build up strong Q2DW signs in the MLT. One reason is likely associated

with a strengthening of the summer easterly jet in the extratropical upper mesosphere favoring a non-linear interaction with the migrating diurnal tide (McCormack et al., 2010). The mesospheric easterly jet itself undergoes a not insignificant variability throughout the years, mainly due to the variation of gravity wave activity as reported in Ern et al. (2013). These circumstances imply the overall complex interactions related to Q2DW activity.

The Q2DW has been studied for decades via ground-based and space-born observations (e.g. Lima et al., 2004; Limpasuvan

et al., 2005; Tunbridge et al., 2011; Gu et al., 2013). All of these techniques have their individual advantages and disadvantages. Analysis from satellites are required to get a global view of the Q2DW activity. Compared with ground-based techniques the temporal resolution of local observations is poor for satellites. To perform long-term studies of e.g. the inter-annual variability of the Q2DW ground-based measurement sites can provide an excellent source of data. Moreover a high temporal resolution offers the possibility of investigating poorly understood non-linear wave wave interactions between Q2DW and atmospheric

waves with even shorter periods, like diurnal or semi-diurnal tides. Both observation types, global and local, complement each other and are required to study the Q2DW in all its facets within the Earth's atmosphere.

One main temporal feature of the quasi 2-day wave is its appearance in burst-like events, meaning that the amplitude strength is highly discontinuous in time. As shown in other studies (Harris and Vincent, 1993; McCormack et al., 2014; Tschanz and Kämpfer, 2015) and in our presented results (Sect. 3), the Q2DW signatures can manifest in a high degree of inter-annual as

well as intra-seasonal variability.

Apart from wind measurements as a proxy for dynamical patterns in the middle atmosphere, it is common to use dynamical tracer observations such as water vapor. In the mesosphere $H_2O$ is photochemically stable for weeks (Brasseur and Solomon, 2006) and this circumstance is used to investigate middle atmospheric wave dynamics from ground-based observations (Scheiben et al., 2014; Tschanz and Kämpfer, 2015; Lainer et al., 2016, 2017). In this study we present quite continuous

observations of the Q2DW signature in middle atmospheric water vapor for 7 years, respectively 84 months, by the middle atmospheric water vapor radiometer MIAWARA at Bern/Zimmerwald ($46.88° \, N$, $7.46° \, E$). Such investigations, especially in the mid-latitudes, are rare and will provide new insights of the Q2DW variability at mesospheric altitudes. Section 2 is dedicated to water vapor radiometric measurements in the middle atmosphere and the corresponding millimeter wave radiometer MIAWARA. Further the $H_2O$ data set of MIAWARA underlain this study is presented. Section 3 focuses on the most impor-

tant results and observed features of the Q2DW above the location of Bern. In particular we put the focus on three sub-areas,





which include climatological features such as averaged monthly mean Q2DW amplitudes, the temporal evolution and observed variability and some explored features indicating non-linear wave-wave interactions based on an auto bicoherence analysis. A conclusion is given in Sect. 4.

## 2   Data from ground-based water vapor radiometry

Ground-based microwave radiometry offers a technique to continuously measure the amount of atmospheric trace gases, such as water vapor, at altitudes between roughly 30 and 80 km under most environmental conditions. Measurements are possible during day, night and under cloudy conditions. As demonstrated by Kämpfer et al. (2012), microwave radiometry is widely used to study the middle atmosphere.

The middle atmospheric water vapor radiometer MIAWARA was built in 2002 at the University of Bern (Deuber et al.,
2004). The Front-End of the radiometer receives emission from the pressure broadened rotational transition line of the $H_2O$ molecule at the center frequency of 22.235 GHz. The retrieval of water vapor from the integrated raw spectra is based on the optimal estimation method (OEM) as presented in Rodgers (2000). We use the ARTS/QPACK software (Eriksson et al., 2005, 2011), where the OEM is used to perform the inversion of the atmospheric radiative transfer model ARTS. A Fast-Fourier Transform (FFT) spectrometer analyses the received microwave signals. The FFT has a spectral resolution of 60 kHz and the
retrieval makes use of an overall spectrum bandwidth of 80 MHz around the center frequency. A monthly mean zonal mean Aura MLS climatology provides the a priori water vapor profile and additionally Aura MLS is used to set the pressure, temperature and geopotential height in the retrieval part. MIAWARA is part of NDACC (Network for the Detection of Atmospheric Composition Change) and is continuously probing middle atmospheric $H_2O$ from the Atmospheric Remote Sensing observatory in Zimmerwald (46.88°N, 7.46°E, 907 m a.s.l.) close to Bern since 2006. In the stratosphere the vertical resolution of
the water vapor profiles is 11 km and degrades to about 14 km in the mesosphere (Deuber et al., 2005). A recent validation against the Aura MLS v4.2 water vapor product (Livesey et al., 2015) revealed that for most months and altitudes the relative differences between MIAWARA and Aura MLS are below 5 % (Lainer et al., 2016). The MIAWARA water vapor data set used during this study has a temporal resolution of 6 hours. This is useful to study not only the Q2DW but also possible interactions with waves of shorter periods like tides. Compared to an even higher temporal resolved $H_2O$ data set like the one used in
Lainer et al. (2017) with a 3 hour time interval, the 6 hours ensure an usability also during summer when the measurement sensitivity is lower.

The MIAWARA $H_2O$ time series between October 2010 and September 2017 is shown in Fig. 1. The corresponding measurement response of 50 % is marked by the white horizontal lines and represents a typical value to where the data can be considered as reliable in regard of the sensitivity to the a priori profile. The measurement response can be obtained from the
retrieval averaging kernel matrix (Rodgers, 2000). Based on the variability of the measurement response, we consider different upper measurement limits between the pressure level range of 0.02–0.05 hPa dependent on the actual month for the whole $H_2O$ data set of 7 years. The approach of the numerical data analyses is explained in the upcoming section.



## 3    Quasi 2-day wave activity

The spectral decomposition of the water vapor measurement time series uses a wavelet-like approach as explained in Studer et al. (2012). In particular, a digital band-pass filter (non-recursive finite impulse response) with a comprised Hamming window with a size of 3 times the central period setting (35–65 h), is applied to the data time series. The $H_2O$ measurements on each retrieval pressure level are handled as distinct data time series. The application of a windowing method to individual measurement time series ensures that the data endpoints fit together. Thus a smoothing out of short-term data fluctuations is characteristic and ensures a good mapping of oscillations with longer periods. Overall the spectral leakage can be reduced by using numerical windowing methods (Harris, 1978). We define the absolute amplitude of the wave as peak-to-peak of the filtered signal and the relative amplitude as relative to the time averaged amount of water vapor measured at the respective pressure level.

Beyond observations of summertime Q2DWs, high Q2DW activity near winter solstices have occasionally been reported from high latitude observations (Nozawa et al., 2003; Tunbridge and Mitchell, 2009; Tschanz and Kämpfer, 2015). A recent study by Madhavi et al. (2015) analyzed COSMIC (Constellation Observing System for Meteorology, Ionosphere, and Climate) GPS RO (Radio Occultation) measurements at mid and high latitudes in regard of the seasonal, latitudinal, and interannual variability of the westward-propagating Q2DW in temperature fields. They found pronounced oscillations with monthly mean amplitudes up to about 8 K during NH fall and winter season in the altitude range 20–60 km. It is of particular interest to investigate Q2DWs observations through the whole year. Scientific reports about Q2DW activity at mid-latitudes in both winter and summer mesospheric conditions are sparse and our study contributes with such observations.

The amplitude analysis of the Q2DW in the period range 43–53 h of our $H_2O$ data set is shown in Fig. 2. The plot is only drawn where the $H_2O$ data is considered as reliable by means of measurement response values greater than 50 %. Both the absolute and relative Q2DW amplitudes are shown. From the overall view, the Q2DW activity is stronger above 52 km (0.5 hPa) altitude than below and shows a highly developed temporal variability. Nevertheless a regular yearly cycle of the 2-day oscillation signatures in water vapor, which is a recurring feature over the 7 investigated years, can be identified. For the location of Bern there is a clear enhancement of Q2DW activity during winter observable. During the summer months we find also Q2DWs but not as pronounced as during winter. This might be related to the lower measurement limit of the $H_2O$ radiometer in summer (see Fig. 1). Typical altitudes where the W3 or W4 Q2DW summer activity maximizes at mid-latitudes (like Bern) is at altitudes above 80 km (McCormack et al., 2014), where our instrument is not capable to retrieve information.

The appearance of the Q2DW can be described as burst-like events that rapidly emerge. The highest amplitude in our data set reached 0.8 ppm (14.4 %) in late January 2015 (around 2015-01-25) peaking at around 0.1 hPa and could be related to a minor sudden stratospheric warming (SSW) event in early January. Manney et al. (2015) reported about this event which had a large impact on transport and the chemical composition of the lower stratosphere in the following weeks and months. Another recurring feature of the wintertime Q2DW over Bern is not only the prevailing high amplitudes in the upper mesosphere but also an activity across all altitude levels down to the stratopause level (1–2 hPa). Another event with Q2DW amplitudes as high as 0.77 ppm (14.6 %) took place end of November 2016 at pressure levels above 0.1 hPa. As it can be seen in Fig. 2, Q2DWs



are not very persistent in time and single burst-like events only last for a couple of days to two weeks at most. From local profile observations of water vapor alone the direction of wave propagation, horizontal and vertical wave lengths or zonal wave numbers, cannot be derived. Therefore additional simultaneous measurements of at least meridional and zonal wind would be required. An excellent possibility in regard of deriving such informations are global re-analysis models.

Pancheva et al. (2016) looked into the global distribution and variability of the Q2DW in the NOGAPS (Navy Operational Global Atmospheric Prediction System)-ALPHA reanalysis model. At middle and high latitudes two different types of waves could be identified: (1) eastward traveling waves with zonal wave numbers 2 and 3 (E2, E3) during winter time and (2) westward traveling waves with zonal wave numbers 2, 3 and 4 (W2, W3, W4) predominantly during the summer months. The same model system was recently used to study a non-linear interaction between the migrating diurnal tide and the W2/W3

waves (Lieberman et al., 2017). The outcome of this interplay of wave forces is split into a westward traveling wave component W4 with a period of around $16\,\mathrm{h}$ and an eastward traveling wave component E2 with a period of 2 days. The W4 wave shows the largest amplitudes in the mid-latitude winter mesosphere and reminds one at a inertia-gravity wave in its behavior. We reported about possible W4 wave observations with a period close to $18\,\mathrm{h}$ in one of our previous papers (Lainer et al., 2017). In Sect. 3.3 we show 4 examples of autobicoherence spectra calculated from 2 months of MIAWARA $H_2O$ data. With this

approach we intend to reveal non-linear wave-wave couplings and show the complexity of middle atmospheric water vapor dynamics.

### 3.1   Monthly climatological overview

The spectral decomposition of 7 years of mesospheric $H_2O$ offers a climatological view of Q2DW activity. Overall 84 months are available to calculate monthly mean wave spectra. Some of those are presented in Figures 3 and 4. For simplicity, we only

present 21 plots per figure focusing on three winter months (December, January, February), respectively summer months (June, July, August), what gives in total 42 monthly mean wave spectra of Q2DW amplitudes within the period range 35–65 hours. By comparing Figures 3 and 4 it is important to take note of the different color bar scales. During the summer months the monthly mean water vapor amplitude maxima do not exceed $0.2$–$0.25\,\mathrm{ppm}$, but during the winter months these values can be higher by $0.1\,\mathrm{ppm}$ ($\sim 50\,\%$).

Overall, a high variability of Q2DW periods from one month to another and from year to year is found for the three summer and winter months. By comparing December, January and February a preference of stronger quasi 2-day wave amplitudes can be attributed to January and February, except for the year 2017. The selected summer months (June, July and August) show an indifferent situation with no obvious preference of stronger Q2DW activity. Relatively strong events occurred in July and August 2011, June 2013 or June 2017. The $H_2O$ amplitudes exceeded $0.2\,\mathrm{ppm}$ and the central periods of maximal Q2DW

mean amplitudes are found between $38$ and $50$ hours. In several Januaries and Februaries between 2010/2011 and 2016/2017 mean Q2DW amplitudes manifest in much higher values above $0.3\,\mathrm{ppm}$ (Feb 2012, Jan 2013, Feb 2013 or Feb 2016) with periods between $40$–$52$ hours. The altitude region where the highest Q2DW amplitudes can be found in all investigated months is somewhere above the stratopause level ($1\,\mathrm{hPa}$). Some monthly averaged $H_2O$ Q2DW spectra have an interesting feature. At a certain altitude range two different period modes of Q2DWs with rather low (close to $36\,\mathrm{h}$) and high (higher than $60\,\mathrm{h}$)



periods are present. Examples in Fig. 3 include August 2013, July 2014 and June 2016. In Fig. 4 such a feature is observable (on a monthly perspective) in January 2011, December 2012 and February 2017. Wave periods close to 36 hours (harmonics of the semi-diurnal tide) are not considered to be within the Q2DW spectrum. The same pertains for wave periods beyond 64 hours, where an influence of ultra-fast Kelvin waves with periods in the range 3–5 days (England et al., 2012) cannot be
excluded. In our data analysis a clear 3-day wave signature is seen for example in November 2010 and 2011.

Averaging Q2DW spectra over all 7 Januaries (Fig. 5a), for instance leads to a similar signature of high amplitudes at the lower and upper branch of Q2DW periods at 0.03–0.04 hPa. Figure 5 clearly shows independently on a certain period band like for Fig. 2, where the Q2DW was constraint to 43–53 hours, that for a typical mid-latitude observation site as Bern strong quasi 2-day oscillations preferably develop during winter months (October to March) rather then summer months (April to
September). The most sharp and distinct Q2DW periods are found during February, October and to some extent also December (Figures 5b, 5j and 5l), meaning that the frequency variability of the wave oscillations is much lower than for example during January, March or November where a horizontal amplitude band indicates a quite high variability (Figures 5a, 5c and 5k). The climatology for December (Fig. 5l) reveals as the only month two peaks of quasi 2-day wave activity at different altitude regions (0.02–0.03 hPa and 0.1–0.2 hPa) with periods near 38 hours. Especially December 2016 has such a pronounced Q2DW
signature as seen in the first subplot of the last row in Fig. 4. The vertical distance between the two wave maxima is about 11 km and the structure could be related to the vertical propagation of planetary waves, what Q2DWs are. The derivation of wave propagation characteristics would require additional observations of wind or the study of model data that could represent the dynamics of water vapor as we observe it with our instrument. From Fig. 5 we get the core message of when it is most likely to see strong Q2DW activity up to altitudes of 70 km, respectively 0.05 hpa (summer) and 75 km, respectively 0.02 hPa
(winter) and this could be relevant to other measurement campaigns aiming at measuring quasi 2-day wave oscillations in the mid-latitude MLT.

A view from a different perspective can be obtained with the histogram plot provided in Fig. 6. There the periods of localized primary and secondary Q2DW events (observed in a monthly mean wave spectrum, as in Figures 3 and 4) are binned and color separated by season. Summer is shown in red and winter in blue colored bars. A primary Q2DW is characterized in
our definition as the wave with the strongest amplitude in the altitude versus period wave spectrum. Beside that one or more secondary Q2DWs can be present with different periods and/or occurrence at other pressure levels. Both primary and secondary Q2DWs have to exceed 0.15 ppm to enter the histogram statistics. The pressure range where the amplitude peaks are valid is set between 0.02 and 2 hPa. The classical 2-day wave periods (50–52 h) count 18 cases out of 110 and show a predominance during winter. The largest amount of Q2DWs have periods in the range 38–40 h (15 in winter, 10 in summer). In total about
20 % of all 110 identified Q2DWs fall into the first bin. Regarding the normal Rossby wave mode W3 with central periods between 50–52 h (Tunbridge et al., 2011) we find a corresponding local maximum of events. The remaining wave periods are ambiguously spread between summer and winter months.

Gu et al. (2013) analyzed the Q2DW behavior for 16 Januaries and Julies in the zonal and meridional wind obtained from the medium frequency radar at Kauai (Hawaii, $22°N$, $160°W$). For January they find most Q2DW periods at 48 h in case of
the meridional wind or 48 h and 51 h in case of the zonal wind. Below 42 and above 54 h no periods were detected that could





be attributed to a Q2DW. A slight displacement towards shorter periods in July is recognizable in their histogram data. Most wind oscillations have either 46 hour (meridional) or 43 hour (zonal) periods.

Afterwards (Sect. 3.2) we restrict the analyses to pressure layer averaged data products and focus more specifically on the monthly mean temporal development of Q2DWs for the whole studied time period of 84 months (7 years) in the resolved

period spectrum between 38 and 64 hours.

### 3.2   Temporal evolution and variability

From the histogram plot (Fig. 6) we got an overview of the distribution of Q2DW periods where the $H_2O$ amplitudes peaked. However it is not less interesting to see how the Q2DW periods evolve in time. In order to emphasize the temporal development we came up with an amplitude matrix plot (Fig. 7) presenting Q2DW period versus time on monthly steps. For both sub figures

different pressure layers are defined where the monthly mean Q2DW amplitudes are further averaged. Figure 7a represents the pressure layer from 0.05–0.2 hPa whereas Fig. 7b covers the data from 0.2–1 hPa. The layer depths in terms of spatial dimensions are 9.6 km, respectively 11.2 km. In both pressure layers a yearly cycle of enhanced quasi 2-day oscillations of various periods is apparent. So to say, the two plots complement the analysis provided with Fig. 2 where only the mean Q2DW (43–53 h) evolution is shown. Now we focus on the hourly resolved monthly mean amplitude of the respective Q2DW

frequency. Higher amplitudes are found towards shorter periods in the summer months in the upper pressure layer, what is consistent to Fig. 6. In general the upper pressure layer is the one where the Q2DW oscillations are more pronounced. Occasionally, the lower pressure layer shows monthly mean $H_2O$ amplitudes slightly higher than 0.2 ppm (December 2012, February 2015), but values as high as 0.3 ppm are never arising as they do at the higher investigated pressure layer. The wintertime maxima of mean amplitudes has a quite high frequency variability. The strongest events exhibit periods above 50

hours (January and February 2015) in the upper mesosphere (Fig. 7a). Two blue columns (May 2011 and May 2017) in each plot are the consequence of larger measurement gaps of the MIAWARA instrument.

The last graphical display (Fig. 8) of this section highlights the temporal development of monthly averaged Q2DW amplitudes (43–53 h) separately for each month of the year. This results in 7 data points per month according to the length of the data set, which is 7 years. The months are distinguished by the given color code. A pressure layer averaging is applied in agreement

with the data presented in the previous Figure 7. January and February 2015 reveal the highest amplitudes of the 43–53 h quasi 2-day wave within our water vapor data set. The amplitudes reach values around 0.27 ppm in the upper mesospheric pressure layer. These high monthly means could be related to SSW dynamics and an enhanced gravity wave activity. Another possibility could be a signature of the maximum phase of the 11-year solar cycle 24. For example Gu et al. (2013) showed that the January Q2DW in zonal and meridional wind has an in phase behavior to the solar irradiance with a leading solar maximum of about

1 year. In the region above the stratopause (lower mesosphere) only February 2015 shows a significant peak, related to the surrounding years (Fig. 8b). We find no clear trend in the temporal evolution of Q2DW activity within the two pressure layers. As it was the outcome before, the winter months tend to have the highest monthly mean quasi 2-day wave $H_2O$ amplitudes and all months indicate higher Q2DW activity in the upper investigated pressure layer between 0.05–0.2 hPa.





### 3.3 Autobicoherence anaylsis

With a bicoherence analysis a wave coupling between two or three waves can be determined. The degree of local quadratic non-linearity gets high when the phase between the waves at a period $s_1$ and $s_2$ (two wave example) is nearly constant over a significant number of realizations. A two wave bicoherence analysis is used to estimate the contribution of second-order
non-linearities to the power of two frequencies (bifrequencies), respectively periods.

In a two-dimensional bicoherence graph as presented in Fig. 9 one usually finds two types of structures: Localized point-like or elongated line-like areas stretching over a bunch of frequencies. First ones indicate sharply defined and locked frequencies, while the latter are likely due to a single frequency mode interacting with a broader range of different frequencies (van Milligen et al., 1995). The peaks in general represent the phase coupling between different wave periods. A significant peak located near
the diagonal slice of the spectrum indicates a phase coupling of the primary frequency mode with its harmonic. Monte Carlo simulations are used to find regions of normalized wavelet power in the autobicoherence spectrum that are significant with respect to a selected confidence interval. In our case a confidence interval of $80\%$ is applied with a total number of 100 iterations within the Monte Carlo simulations. An in depth view about methodical and computational details of the autobicoherence analysis is given in Schulte (2016) and Grinsted et al. (2004).
Figure 9 presents four autobicoherence spectra from two months of pressure layer averaged MIAWARA water vapor time series. Figure 9a and Fig. 9b focus on January and February 2016, while Fig. 9c and Fig. 9d show results for November and December 2016. In case of January and February 2016 significant phase coupling can be found between a quasi $18\,h$ ($16-18\,h$) wave and the Q2DW with a period slightly below $48\,h$ (Fig. 9a, label **B**) in the lower pressure layer and a coupling of $18\,h$ oscillations to diurnal periodicities in the upper pressure layer (Fig. 9b, label **D**). Between $0.2$ and $1\,hPa$ the diurnal tide is
to high degree (power: $0.8$) phase coupled to the semi-diurnal tide ($12\,h$ period), as the red area at coordinate point (24, 24) shows (Fig. 9a, label **C**). In the upper mesosphere this tidal wave behavior is lost, but here a tidal period $s_1$ manifests in a line-like area across $s_2$ periods (not significant within the $80\%$ confidence interval) in the Q2DW period range below 48 to above 64 hours (Fig. 9b, label **E**). In Fig. 9a the highest wavelet power (label **A**) is found at coordinates $(48, >64)$ and could be related to an interference of the Q2DW with the quasi $18\,h$ wave which itself is likely to originate from a non-linear wave-
wave coupling between the diurnal tide and the westward traveling quasi 2-day wave (W2) (Lieberman et al., 2017). A recent study by Lainer et al. (2016) revealed dominant oscillations in mesospheric water vapor profiles with a period close to $18\,h$ in Northern Hemispheric winter months. However such oscillations within sub-diurnal period spectrum in the MLT can also be related to low frequency inertia-gravity waves, as shown by Li et al. (2007) with measurements from a sodium lidar system over Fort Collins, Colorado ($41°N$, $105°W$).
The MIAWARA autobicoherence spectra for November and December show for both altitude regions similar quadratic phase coupling signatures. High common wavelet power is found between $18$ and $32$ hours (Figs. 9c and d, labels **F** and **I**). The red spot (label **H**) at coordinate (32, 32) indicates also a coupling between the harmonic of the $32\,h$ oscillations and the primary frequency. At $s_1$ Q2DW periods have a significant phase coupling to even longer periods (up to $80\,h$) as it can be seen in Figure 9c near label **G**. Even though we only made use of a single mesospheric $H_2O$ data set, atmospheric wave patterns and





interactions can be studied. Evidences were found that wave-wave interactions bewteen Q2DWs, diurnal tides and quasi $18\,\mathrm{h}$ waves occur in the winter mid-latitude mesosphere shown by high non-linear phase couplings in the autobicoherence spectra of MIAWARA $H_2O$ data.

## 4    Conclusions

The study of quasi 2-day planetary waves in the MLT is of importance to improve the understanding of the Earth's atmosphere. The dissipation of atmospheric waves in the MLT induces rapid changes on the background dynamics, which in turn affects the composition of the atmosphere through turbulent mixing or the general alternating of the circulation. This Q2DW-driven variability can be seen in long-living trace gases like water vapor.

The MIAWARA radiometer provides reliable, long-term observations of middle atmospheric water vapor since 2007. Here we made use of data since October 2010 right after the instrument was essentially improved by a hardware update resulting in shorter integration times of the $22\,\mathrm{GHz}$ $H_2O$ spectra and thus a higher temporal resolution. A temporal data resolution of 6 hours was the starting point for the long-term analyses of Q2DW activity above the stratopause up to an altitude of $75\,\mathrm{km}$ ($0.02\,\mathrm{hPa}$) during winter and $70\,\mathrm{km}$ ($0.05\,\mathrm{hPa}$) during summer months when the increase of atmospheric opacity reduces the upper measurement limit.

Our key results regarding the long-term Q2DW behavior above the mid-latitude observation site at Bern are briefly summarized:

– Q2DW ($43$–$53\,\mathrm{h}$) activity as observed by MIAWARA $H_2O$ profiles is strongest in the upper mesosphere and during winter months and emerges in burst-like events. We note the altitude limitation of the MIAWARA instrument during summer which is limited to about $70\,\mathrm{km}$.

– Highest individual Q2DW amplitudes reach $0.8\,\mathrm{ppm}$ and are likely related to SSW activity

– Monthly mean Q2DW amplitude spectra show a broad variability of periods between $38$ and $64\,\mathrm{h}$

– A monthly climatological overview for 7 years indicates that in January, February and November the amplitude peaks of Q2DWs are highest (up to $0.3\,\mathrm{ppm}$) in the observed altitude region

– A significant fraction (about $20\,\%$) of observed Q2DW events in summer and winter are manifesting periods between $38$ and $40\,\mathrm{h}$

– The evolution of different Q2DW periods (monthly average) over 84 months revealed a yearly signature of enhanced wave activity during winter months

– Non-linear quadratic phase coupling detected between Q2DW, diurnal and quasi $18\,\mathrm{h}$ $H_2O$ oscillations

We showed that measurements from ground-based microwave radiometers can be used to assess the quasi 2-day wave activity at local observation sites. Even if data sets from satellite measurement platforms like Aura MLS (operational since July 2004)



can provide a global perspective of Q2DWs, observations from ground can be used for validation purposes and more important for long-term monitoring of wave activity and in case of Q2DWs they can capture the interaction with shorter periodical waves like tides or semi-diurnal oscillations, which are much harder to resolve (temporally) by satellites (Nyquist sampling problem).

*Data availability.* Data from the ground-based microwave radiometer MIAWARA is publicly available from the NDACC database as
5   monthly files with a diurnal temporal resolution (ftp://ftp.cpc.ncep.noaa.gov/ndacc/station/bern). Data with a higher temporal resolution is only available upon request.

*Competing interests.* Hereinafter all authors declare to have no competing interests.

*Acknowledgements.* This work is supported by Swiss National Science Foundation Grant 200020-160048 and MeteoSwiss in the frame of the GAW project "Fundamental GAW parameters measured by microwave radiometry". Further, we thank Aslak Grinsted for providing
10   wavelet-coherence software (http://www.glaciology.net/wavelet-coherence).



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





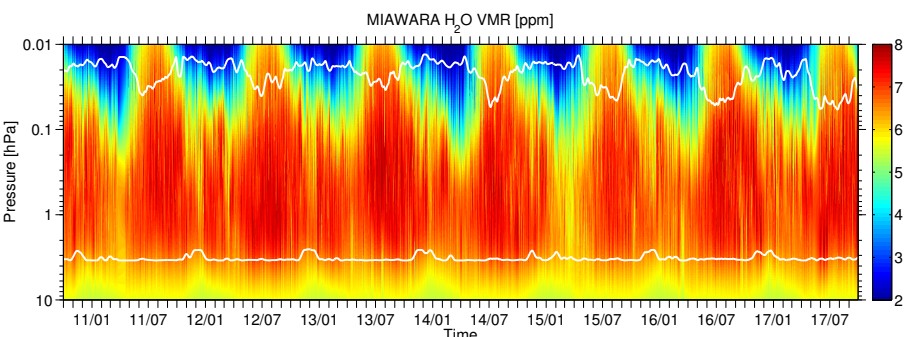

**Figure 1.** Water vapor volume mixing ratio [ppm] time series as measured by the MIAWARA microwave radiometer between October 2010 and September 2017. The horizontal white lines indicate at which pressure levels the measurement response drops below $50\%$. Clearly seen is the annual cycle in the mesosphere with a $H_2O$ maximum in summer. The measurement response is affected by tropospheric opacity which is higher in summer what leads to the observed variation in time.





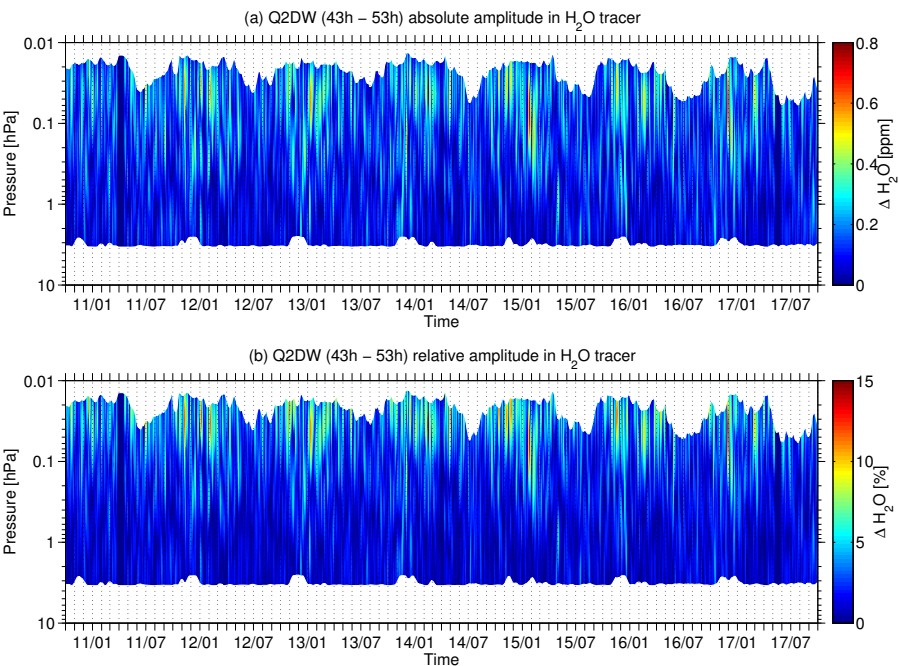

**Figure 2.** Evolution of absolute (a) and relative (b) Q2DW amplitude in water vapor data from the middle atmospheric water vapor radiometer MIAWARA in the time period from 1 October 2010 to 30 September 2017. The data product is shown in the altitude region where it can be regarded as reliable according to Fig. 1.



**Figure 3.** Monthly averaged water vapor wave amplitude spectra with periods between 35 and 65 hours in units of [ppm]. Presented are the months June (first column), July (second column) and August (third column) for the years from 2011 to 2017.





**Figure 4.** Same representation as in Fig. 4, but for the months December (first column), January (second column) and February (third column) between 2010 and 2016, 2011 and 2017.





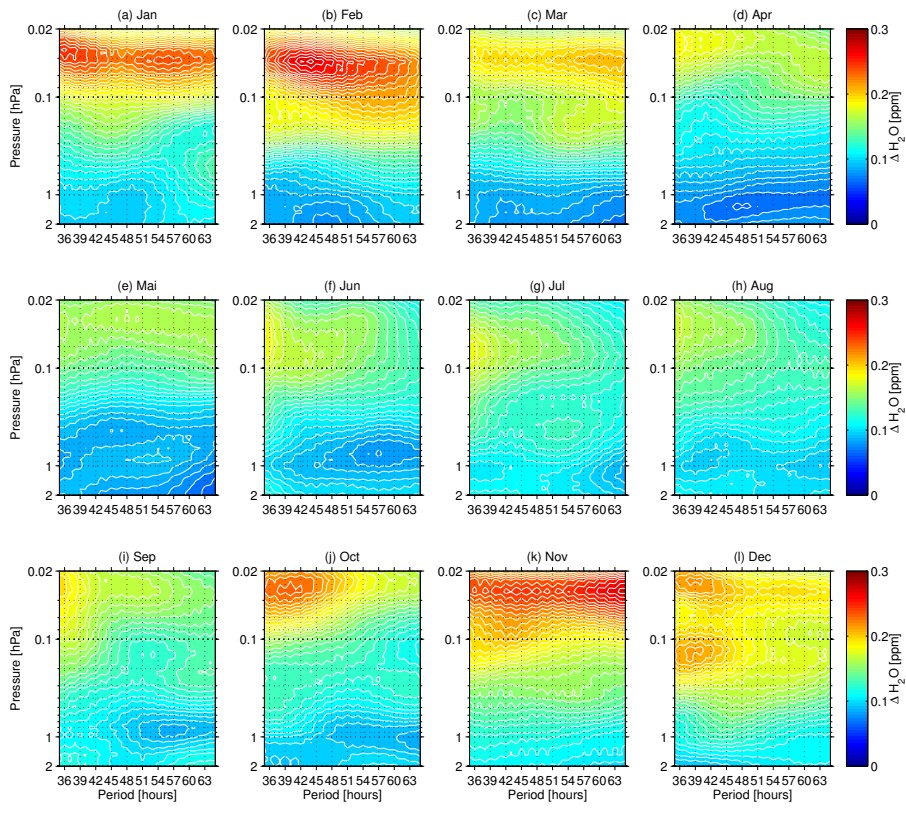

**Figure 5.** Monthly climatology (January to December, (a)–(l)) of wave amplitude spectra for periods between 35 and 65 hours over a period of 7 years derived from MIAWAWA $H_2O$ data. The covered altitude range in terms of pressure levels goes from 2 to $0.01\,\mathrm{hPa}$.

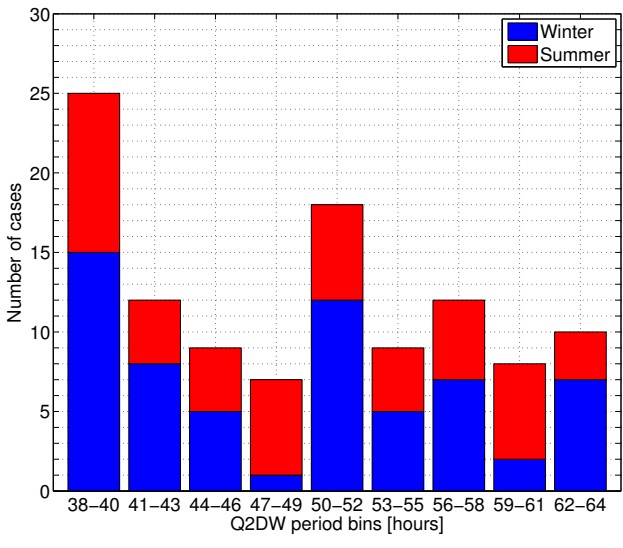

**Figure 6.** Histogram of Q2DW periods observed with the MIAWARA water vapor radiometer. Shown are the number of cases versus period bins with 3 h width in which Q2DW events could be identified. The selected criterion of a Q2DW event was a localized maximum in a monthly averaged $H_2O$ wave spectrum exceeding 0.15 ppm. The bar plots are stacked which divides them into winter (blue) and summer (red) groups.



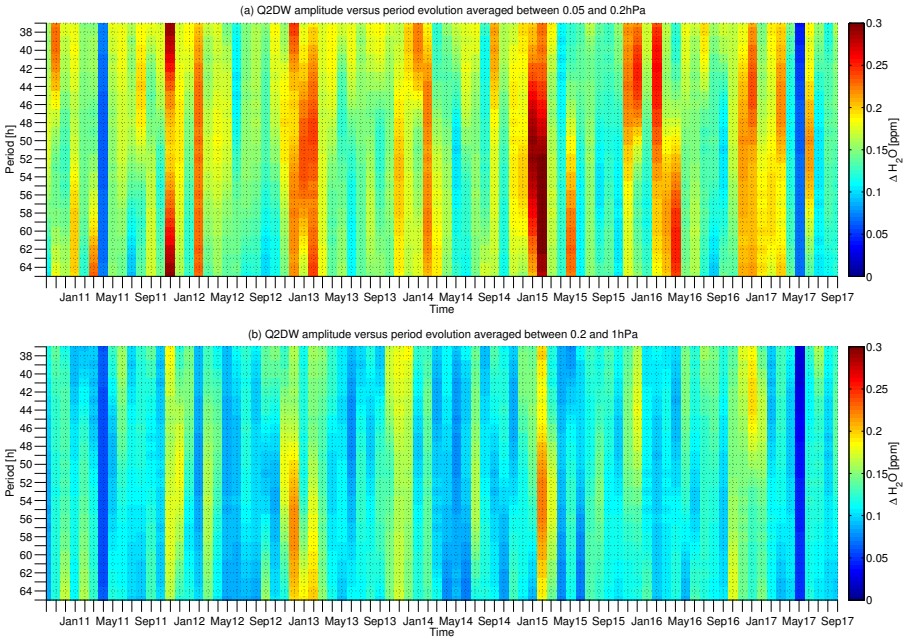

**Figure 7.** Matrix plots of the temporal evolution of monthly mean Q2DW amplitudes in units of [ppm] in dependency on the period (38–64 h). The top plot shows the pressure layer averaged wave amplitudes between 0.05–0.2 hPa, while the bottom plot does between 0.2–1 hPa.





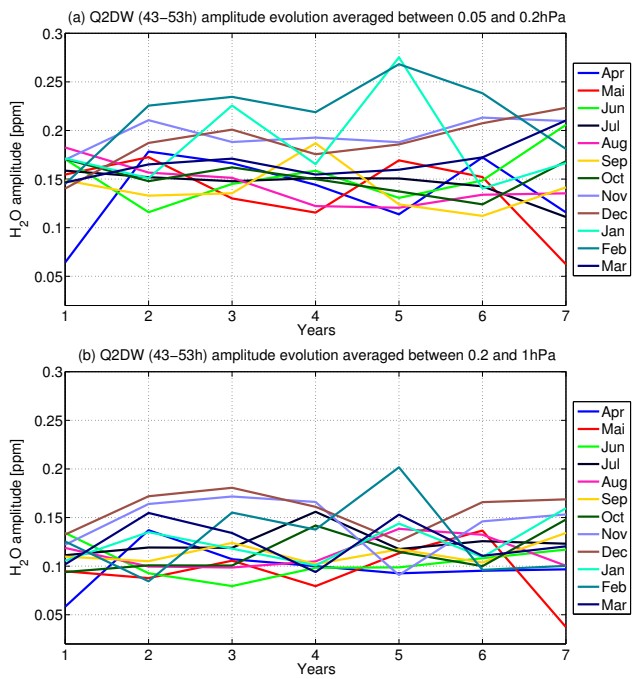

**Figure 8.** Monthly break down of Q2DW (43–53 h) amplitude development over the 7 investigated years. The $H_2O$ amplitudes are pressure layer averaged: 0.05–0.2 hPa (top plot) and 0.2–1 hPa (bottom plot).





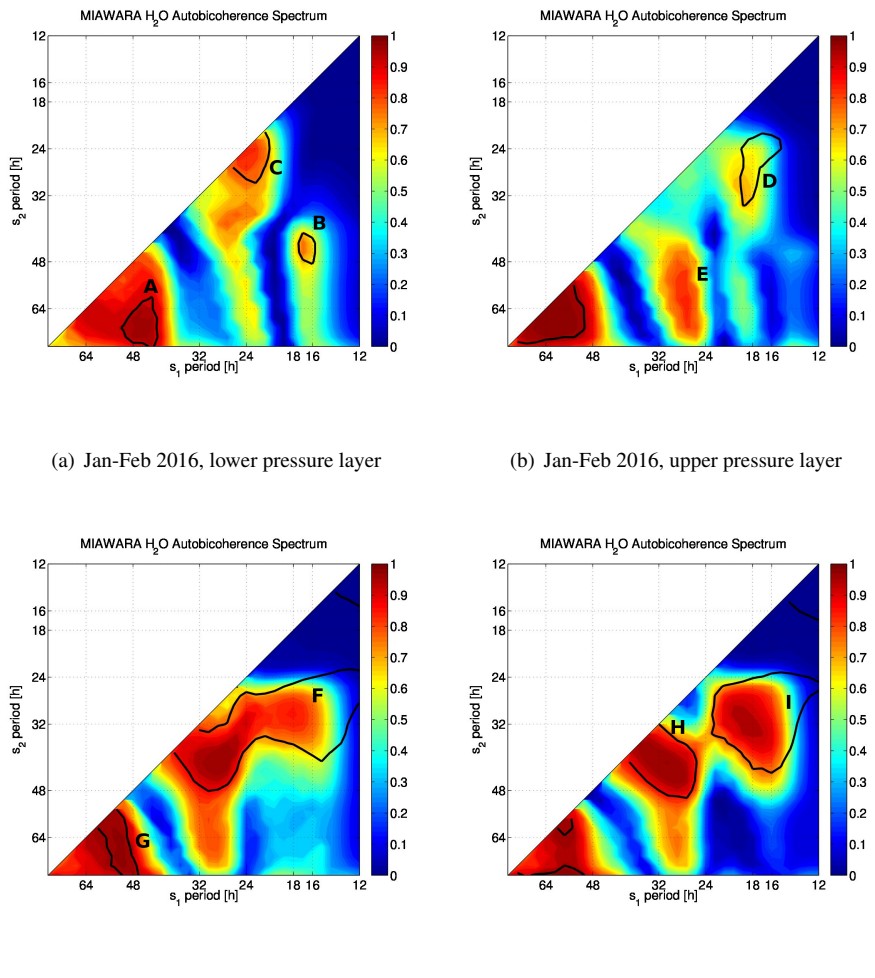

(a) Jan-Feb 2016, lower pressure layer  (b) Jan-Feb 2016, upper pressure layer

(c) Nov-Dec 2016, lower pressure layer  (d) Nov-Dec 2016, upper pressure layer

**Figure 9.** Wavelet-based autobicoherence spectrum from pressure layer averaged MIAWARA water vapor time series with individual lengths of two months. The chosen pressure layers are $0.05$–$0.2\,\mathrm{hPa}$ and $0.2$–$1\,\mathrm{hPa}$. Thick contours enclose regions of $80\,\%$ point-wise confidence after controlling the FDR (False Detection Rate). The diagonal line separates the 2-dimensional spectrum into two symmetric regions. Interesting features on the plots are labeled with capital letters (**A**–**H**).