# Peer review of "Long-term observation of mid-latitude quasi 2-day waves by a water vapor radiometer"

_Atmospheric Chemistry and Physics, 2017_

## Referee Comment (RC1) · Anonymous Referee #1 · 8 Feb 2018

The quasi 2-day waves are one of the most extensively observed planetary waves by different ground based and satellite instruments. In this way the most prominent features of the global space distribution and seasonal and intra-seasonal variabilities of these planetary waves have been already known. Most of these features are also well numerically simulated. This paper presents long-term observations of the quasi 2-day waves by a water vapor radiometer at a mid latitude station Bern. The use of water vapor for studying the planetary waves in the middle atmosphere is not a new practice; there are several reports for different planetary waves, as Nielsen et al. (JGR, 2010), Scheiben at al. (ACP, 2014), etc. and particularly for the quasi 2-day waves for example, McCormack et al. (JGR, 2009). There are only a few studies on the interannual variability of the quasi 2-day waves which however have not been able to present con-

vincing results mainly because of not enough long time observations. According to the title of the present paper we expected its main contribution to be namely in clarifying the interannual variability of these waves. However the use of only seven years (October 2010 – September 2017) of water vapor radiometer measurements is shorter time interval than previously used measurements (as for example, Huang et al. (2013) used 10 years of SABER temperature data) and definitely not enough for considering the interannual variability. The use of only single station measurements significantly limits the ability for studying the spatial structures of these waves; only their vertical structure could be considered. The authors however presented only the vertical structure of the wave amplitudes. There is no any information about the wave phases, i.e. it is not possible to understand if the found waves are vertically propagating or trapped waves. The only convincing result from the data analysis is that the quasi 2-day wave activity is stronger in winter than in summer (probably because the summer observations are limited up to about 70 km height). It is mentioned that the large winter wave amplitudes are likely related to SSW but this issue is not particularly investigated. The bicoherence spectra indicated some nonlinear coupling between the quasi 2-day wave, diurnal tide and quasi 18-hour oscillation, but this is a well known result reported in many papers.

General comment: This study suffers from the lack of new scientific results. If the authors want to add values to these single station measurements they have to combine them with the satellite observations and to make an attempt to clarify if the quasy-2-day waves they observe belong to some of the known modes or are a combination of a few modes. Without such information and the lack of any phase results the present paper shows only the observations of quasi-2-day oscillations, nothing more.

---

## Referee Comment (RC2) · Anonymous Referee #2 · 8 Apr 2018

Radio aeronomy has matured to an established technique for remote sensing of the middle atmosphere. Ground-based measurements of the middle atmosphere will play a very important role in the future since several of the satellite systems, dedicated for middle atmospheric studies, are aging and few replacements are planned.

It has become more and more clear that wave activity in the middle atmosphere can have an impact on the troposphere. This means that the kind of studies, described in this paper, are of utmost importance.

In my opinion the paper is very interesting, well written and easy to follow. My only concern is why the authors do not compare their results with model data like WACCM or datasets like ECMWF. Such a comparison would give additional strength to the paper.

[Figure]

If the authors justify why no comparisons have been made, I consider the article to be published in ACP

---

## Author Comment (AC1) · 3 May 2018

**Long-term observation of mid-latitude quasi 2-day waves by a water vapor radiometer**

*Martin Lainer, Klemens Hocke and Niklaus Kämpfer*
* * *
**Responses to comments on ACPD paper acp-2017-115**
* * *
**Color Code:** Referee comments, Authors response
* * *
We would like to thank all anonymous referees for their comments to our ACPD paper.

Please find our general responses to Reviewers #1 and #2 below.

**1 Response to Referee #1**

The quasi 2-day waves are one of the most extensively observed planetary waves by different ground based and satellite instruments. In this way the most prominent features of the global space distribution and seasonal and intra-seasonal variabilities of these planetary waves have been already known. Most of these features are also well numerically simulated. This paper presents long-term observations of the quasi 2-day waves by a water vapor radiometer at a mid latitude station Bern. The use of water vapor for studying the planetary waves in the middle atmosphere is not a new practice; there are several reports for different planetary waves, as Nielsen et al. (JGR, 2010), Scheiben at al. (ACP, 2014), etc. and particularly for the quasi 2-day waves for example, McCormack et al. (JGR, 2009). There are only a few studies on the interannual variability of the quasi 2-day waves which however have not been able to present convincing results mainly because of not enough long time observations. According to the title of the present paper we expected its main contribution to be namely in clarifying the interannual variability of these waves. However the use of only seven years (October 2010 September 2017) of water vapor radiometer measurements is shorter time interval than previously used measurements (as for example, Huang et al. (2013) used 10 years of SABER temperature data) and definitely not enough for considering the interannual variability. The use of only single station measurements significantly limits the ability for studying the spatial structures of these waves; only their vertical structure could be considered. The authors however presented only the vertical structure of the wave amplitudes. There is no any information about the wave phases, i.e. it is not possible to understand if the found waves are vertically propagating or trapped waves. The only convincing result from the data analysis is that the quasi 2-day wave activity is stronger in winter than in summer (probably because the summer observations are limited up to about 70 km height). It is mentioned that the large winter wave amplitudes are likely related to SSW but this issue is not particularly investigated. The bicoherence spectra

indicated some nonlinear coupling between the quasi 2-day wave, diurnal tide and quasi 18-hour oscillation, but this is a well known result reported in many papers.

General comment: This study suffers from the lack of new scientific results. If the authors want to add values to these single station measurements they have to combine them with the satellite observations and to make an attempt to clarify if the quasi-2-day waves they observe belong to some of the known modes or are a combination of a few modes. Without such information and the lack of any phase results the present paper shows only the observations of quasi-2-day oscillations, nothing more.

- Ground-based observations of quasi 2-day waves are rare and there are no studies on Q2DWs from ground over such a long period of 7 years. The MIAWARA observations rank among the longest records of microwave radiometric $H_2O$ measurements in the strato- and mesosphere from ground. We think that satellite data can be problematic for these short term variabilities due to long reoccurrence times over the same location. To investigate non-linear interactions to tides on a single event basis is not possible with e.g. Aura MLS data.

- We agree that with our $H_2O$ profile data it is not possible to study detailed structures or wave characteristics (e.g. phases, wavenumbers). Due to the low vertical resolution of MIAWARA, which is in the order of $10\,\mathrm{km}$, it is also difficult to detect vertical wave structures.

- The reviewer stated that there are many papers showing nonlinear coupling between the quasi 2-day wave, diurnal tide and quasi 18-hour oscillation. We are aware of some numerical investigations about this (e.g. McCormack et al., 2010; Lieberman et al., 2017) and a few papers on meteor radar observations (e.g. Huang et al., 2013). We would appreciate if the referee could point us to the many papers we are missing. With our measurements we showed that these interactions can be determined also by the use of water vapor data.

**2 Response to Referee #2**

Radio aeronomy has matured to an established technique for remote sensing of the middle atmosphere. Ground-based measurements of the middle atmosphere will play a very important role in the future since several of the satellite systems, dedicated for middle atmospheric studies, are aging and few replacements are planned.

It has become more and more clear that wave activity in the middle atmosphere can have an impact on the troposphere. This means that the kind of studies, described in this paper, are of utmost importance.

In my opinion the paper is very interesting, well written and easy to follow. My only

concern is why the authors do not compare their results with model data like WACCM or datasets like ECMWF. Such a comparison would give additional strength to the paper.

If the authors justify why no comparisons have been made, I consider the article to be published in ACP.

- The reason why we did not compare Q2DW results to ECMWF model data is because in general $H_2O$ volume mixing ratios in the middle atmosphere are badly represented in the model analyses. The discrepancies between measurements and model analyses in the amount of $H_2O$ are up to around $20\,\%$ with an underestimation of the model. In the past we also compared results from WACCM simulations to our observed oscillations in $H_2O$, but in case of the quasi 18-hour oscillation (Lainer et al., 2017) no obvious $18\,h$ variability could be found. That is also the reason why we thought it is not worth to include WACCM in the study of quasi 2-day waves.

**References**

Huang, K. M., Liu, A. Z., Lu, X., Li, Z., Gan, Q., Gong, Y., Huang, C. M., Yi, F., and Zhang, S. D. (2013). Nonlinear coupling between quasi 2day wave and tides based on meteor radar observations at maui. *Journal of Geophysical Research: Atmospheres*, 118(19):10,936–10,943.

Lainer, M., Hocke, K., Rüfenacht, R., and Kämpfer, N. (2017). Quasi 18 h wave activity in ground-based observed mesospheric h₂o over bern, switzerland. *Atmospheric Chemistry and Physics*, 17(24):14905–14917.

Lieberman, R. S., Riggin, D. M., Nguyen, V., Palo, S. E., Siskind, D. E., Mitchell, N. J., Stober, G., Wilhelm, S., and Livesey, N. J. (2017). Global observations of 2 day wave coupling to the diurnal tide in a high-altitude forecast-assimilation system. *Journal of Geophysical Research: Atmospheres*, 122(8):4135–4149. 2016JD025144.

McCormack, J. P., Eckermann, S. D., Hoppel, K. W., and Vincent, R. A. (2010). Amplification of the quasitwo day wave through nonlinear interaction with the migrating diurnal tide. *Geophysical Research Letters*, 37(16).

---

## Referee Report (RR1)

General comments on MS No: **acp-2017-1150-version3** "*Long-term observation of mid-latitude quasi 2-day waves by a water vapor radiometer*" by Martin Lainer, Klemens Hocke, and Niklaus Kämpfer

I am not satisfied by both the response to my comments and the revised version of this paper. As a consequence my general concerns remain the same as before, i.e. **this study suffers from the lack of new scientific results; actually the paper presents only observations of quasi-2-day oscillations, nothing more.**

In order to clarify why I am not satisfied by the response to my comments I am going to comment the authors' response. For convenience, their response is copied here in italics.

*1. Ground-based observations of quasi 2-day waves are rare and there are no studies on Q2DWs from ground over such a long period of 7 years. The MIAWARA observations rank among the longest records of microwave radiometric H2O measurements in the strato- and mesosphere from ground. We think that satellite data can be problematic for these short term variabilities due to long reoccurrence times over the same location. To investigate non-linear interactions to tides on a single event basis is not possible with e.g. Aura MLS data.*

Hardly could agree that ground-based observations of the quasi-2-day waves are rare; please, note that the first report about these waves was presented by Müller in 1972 using meteor radar observations. Later numerous papers based on D1 radio wind measurements, MF and meteor radars, OH temperature measurements, lidars, etc. have been published. Moreover particular multi-instrument observational campaigns were organized for studying the MLT dynamics, including the quasi-2-day waves as well. I cannot agree also that there are not studies based on ground-based measurements with such a long period of 7 years. Please, see below some papers presenting longer than 7 years measurements:

Harris, T. J. (1994), A long-term study of the quasi-two-day wave in the middle atmosphere, J. Atmos. Terr. Phys., 56, 569–579, doi:10.1016/0021-9169(94)90098-1 (**12** years of MF radar measurements 1980-1991)

Lilienthal, F. and Ch. Jacobi (2015), Meteor radar quasi 2-day wave observations over 10 years at Collm (51.3° N, 13.0° E), *Atm. Chem Phys.*, 15, 9917-9927(11), doi:10.5194/acp-15-9917-2015 (**10** years of meteor radar measurements 2004-2013)

Rao, N. V., V. R. Madineni, C. Vedavathi, T. Tsuda, et al. (2016), Seasonal, inter-annual and solar cycle variability of the quasi two day wave in the low-latitude mesosphere and lower thermosphere. *J. Atmos. Sol.-Terr. Phys.*, 152, doi:10.1016/j.jastp.2016.11.005 (**17** years of MF radar 1993-2009 and **10** years of meteor radar measurements 2005-2014)

Offermann, D., P. Hoffmann, P. Knieling, R. Koppmann, J. Oberheide, D. M. Riggin, V. M. Tunbridge, and W. Steinbrecht (2011), Quasi 2 day waves in the summer mesosphere: Triple structure of amplitudes and long-term development, *J. Geophys. Res.*, 116, D00P02, doi:10.1029/2010JD015051 (**15** years OH temperature measurements 1988-1993)

Jacobi, Ch., P. Hoffmann, and D. Kürschner (2008), Trends in MLT region winds and planetary waves, Collm (52°N, 15° E), *Ann. Geophys.*, 26, 1221–1232 (**26** years of D1 radio wind measurements, 1980-2005)

I agree that the satellite studies of planetary waves can be complicated by aliasing but recently Pancheva et al. (2018), investigating the long-term variability of the quasi 2-day waves observed by MLS/Aura, first presented a detailed comparison between the altitude structures of all (eastward and westward travelling) QTDWs extracted from both the synoptic NOGAPS-ALPHA data and the asynoptic MLS/Aura ones to determine the aliasing. Hence that satellite data can be used quite successfully for studying planetary waves.

Sorry, I have never mentioned that the MLS/Aura data can be used for investigating the non-linear interaction between the tides and planetary waves; Aura is sun synchronized satellite and the tides cannot be determined at all.

*2. We agree that with our H2O profile data it is not possible to study detailed structures or wave characteristics (e.g. phases, wavenumbers). Due to the low vertical resolution of MIAWARA, which is in the order of 10 km, it is also difficult to detect vertical wave structures.*

You have single station measurements (i.e. the longitude and latitude structures of the wave cannot be defined) with low vertical resolution that does not allow the vertical wave structure to be determined. As a consequence I mentioned that **the paper presents only observations of quasi-2-day oscillations, nothing more.**

*3. The reviewer stated that there are many papers showing nonlinear coupling between the quasi 2-day wave, diurnal tide and quasi 18-hour oscillation. We are aware of some numerical investigations about this (e.g. McCormack et al., 2010; Lieberman et al., 2017) and a few papers on meteor radar observations (e.g. Huang et al., 2013a). We would appreciate if the referee could point us to the many papers we are missing. With our measurements we showed that these interactions can be determined also by the use of water vapor data.*

Please, see below only part of the papers based on ground-based measurements where the non-linear interaction between tides and quasi-2-day wave is studied. If in the title is mentioned 'planetary waves' this means that besides the ~2-day wave other PW waves are considered as well.

Clark, C. C., and J. S. Bergin (1997), Bispectral analysis of mesosphere winds, *J. Atmos. Terr. Phys.*, 59, 629– 639

Kamalabadi, F., J. Forbes, N. Makarov, and Y. Portnyagin (1997), Evidence for nonlinear coupling of planetary waves and tides in the Antarctic mesopause, *J. Geophys. Res.*, 102(D4), 4437– 4446

Beard, G.A., Mitchell, N.J., Williams, P.J.S., Kunitake, M., (1999), Non-linear interactions between tides and planetary waves resulting in periodic tidal variability. *J. Atmos. Sol.-Terr. Phys.*, 61, 363–376.

Jacobi, Ch., (1999), Non-linear interaction of planetary waves and the semidiurnal tide as seen from midlatitude mesopause region winds measured at Collm, Germany. *Meteorology Zeitschrift*, 8, 28–35.

Pancheva D. (2001), Non-linear interaction of tides and planetary waves in the mesosphere and lower thermosphere: observations over Europe, *Phys. Chem. Earth (Part C)*, v.26, 6, pp. 411-418.

Gurubaran, S., S. Sridharan, T.K. Ramkumar and R. Rajaran (2001), The mesospheric quasi-2-day wave over Tirunelveli (8.7N), *J. Atmos, Sol.-Terr. Phys.*, 10, 975-985, doi: 10.1016/S1364-6826(01)00016-5.

Pancheva, D. (2006), Quasi-2-day wave and tidal variability observed over Ascension Island during January/February 2003, *J. Atmos. Sol.-Terr. Phys.*, 68, 390-407.

Kumar, K. K., V. Deepa, M. Antonita, and G. Ramkumar (2008), Meteor radar observations of short-term tidal variabilities in the low-latitude mesosphere-lower thermosphere: Evidence for nonlinear wave-wave interactions, *J. Geophys. Res.*, 113, D16108, doi:10.1029/2007JD009610.

Suresh Babu, V., K. Kishore Kumar, S. R. John, K. V. Subrahmanyam, and G. Ramkumar (2011), Meteor radar observations of short-term variability of quasi 2 day waves and their interaction with tides and planetary waves in the mesosphere–lower thermosphere region over Thumba (8.5°N, 77°E), *J. Geophys. Res.*, 116, D16121, doi:10.1029/2010JD015390

De Oliveira Alves, E., L. M. Lima et al. (2013), Non-linear interaction between diurnal todal and 2-day wave in the meteor winds observed at Cachoeira Paulista and São João do Carrir, Brazil: A case study, *Rev. Bras. Geofisica*, 31(3), 403-412.

Guharay, A. and P.P. Batista, B.R. Clemesha, (2015), On the variability of the diurnal tide and coupling with planetary waves in the MLT over Cachoeira Paulista (22.7°S, 45°W), *J. Atmos. Sol.-Terr. Phys.*, 133, 7-17, doi: 10.1016/j.jastp.2015.07.016

Finally, I would like to clarify that the sum secondary wave generated by the non-linear coupling between the quasi-2-day wave and the diurnal tide should be **~16-hour wave**; the 18-h wave can be generated between the ~3-day wave and the diurnal tide.

---

## Referee Report (RR2)

The Quasi 2-day wave (QDTW) in the middle atmosphere has been extensively studied in the past, but still there are only few long-term observations of this wave in the winter middle atmosphere from ground-based observations, and additional data are therefore necessary.

The authors present data from 7 years of water vapor radiometer observations. The paper is well written, the method of observation and analysis is well described and the results are clearly presented. I recommend the paper to be published in ACP.

Although the paper may, in my opinion, be published in the present form, the authors may wish to consider/discuss a few points:

The authors are right when explaining that reanalysis water vapor is most probably not accurate enough to analyze the QDTW from that. But the wave should be visible in other parameters, too, and a future study could make use of this.

QDTW phases are not shown, although they might be of interest e.g. to check possible phase locking with the tides.

On Figure 8 the years should be provided on the abscissa

The bicoherence spectra are shown only for 2 months out of 84 available. So the conclusion drawn from them should be made with caution, and actually I would have been interested in what happens during the other months.

---

## Author Response (AR2)

**Long-term observation of mid-latitude quasi 2-day waves by a water vapor radiometer**

*Martin Lainer (on behalf of all co-authors)*
* * *
**Response to comments on ACPD paper acp-2017-115, Minor Revision**
* * *
**Color Code:** Referee comments, Authors response, Changes in manuscript

We would like to thank the two anonymous referees for their comments to our first revised paper version.

Please find our new responses for the minor revision to the Reviewers below, together with some changes to our manuscript. At the end we included a marked-up version of the manuscript showing these new changes.

**1 Response to Referee #3**

The Quasi 2-day wave (QDTW) in the middle atmosphere has been extensively studied in the past, but still there are only few long-term observations of this wave in the winter middle atmosphere from ground-based observations, and additional data are therefore necessary.

The authors present data from 7 years of water vapor radiometer observations. The paper is well written, the method of observation and analysis is well described and the results are clearly presented. I recommend the paper to be published in ACP.

Although the paper may, in my opinion, be published in the present form, the authors may wish to consider/discuss a few points:

The authors are right when explaining that reanalysis water vapor is most probably not accurate enough to analyze the QDTW from that. But the wave should be visible in other parameters, too, and a future study could make use of this.

QDTW phases are not shown, although they might be of interest e.g. to check possible phase locking with the tides.

On Figure 8 the years should be provided on the abscissa

The bicoherence spectra are shown only for 2 months out of 84 available. So the conclusion drawn from them should be made with caution, and actually I would have been

interested in what happens during the other months.

- Yes, the wave should be visible in other parameters like temperature for instance and operational or reanalysis models can be used of course to study this in future investigations. We will make a note in the conclusions about it.

- We agree that the investigation of wave phases could be interesting to check the phase locking behavior between different waves. Unfortunately our developed wave analysis program has not yet an implementation to look at wave phases. Therefore the effort to do this would go beyond a minor revision unfortunately. We also have to note that the first author is no longer affiliated with the University of Bern, which complicates the situation.

- For Figure 8, we did not show the actual years in the x-axis because the beginning of the year is different between the months. For example April would start in 2007 while January in 2008. But we will make a note in the Figure caption about the years.

- With the bicoherence spectra we wanted to demonstrate that we are able to identify non-linear interactions in our data set, especially between the QTDW and tides. So from a large set of months we chose 2 time series of 2 months (Jan/Feb 2016 and Nov/Dec 2016) where those interactions could be identified. The analyses of more months would be definitely interesting, but is beyond this paper.

- Section 4, page 10, line 9ff: Included a statement about the fact that Q2DWs can be studied in other model parameters like temperature with the intention to compare it to ground-based observation results.

- Figure 8: Included in the caption: "The years range dependent on the month from either 2010 to 2016 (October to December) or from 2011 to 2017 (January to September)."

**2 Response to Referee #2**

We first want to mention that we will only respond to a selection of Referee #2 comments, which we believe are relevant at this revision stage.

Hardly could agree that ground-based observations of the quasi-2-day waves are rare; please, note that the first report about these waves was presented by Mller in 1972 using meteor radar observations. Later numerous papers based on D1 radio wind measurements, MF and meteor radars, OH temperature measurements, lidars, etc. have been

published. Moreover particular multi-instrument observational campaigns were organized for studying the MLT dynamics, including the quasi-2-day waves as well. I cannot agree also that there are not studies based on ground-based measurements with such a long period of 7 years. Please, see below some papers presenting longer than 7 years measurements.

- Ok we see that there are studies based mostly on active remote sensing techniques (radar, lidar) in regards of Q2DW observations. We precised our statement to the fact that we believe is true that passive remote sensing observations of Q2DWs can be seen as "rare".

- Included a paper citation, e.g. Lilienthal 2015 in the conclusions. And we connect the "rare" observation expression to passive remote sensing techniques.

I agree that the satellite studies of planetary waves can be complicated by aliasing but recently Pancheva et al. (2018), investigating the long-term variability of the quasi 2-day waves observed by MLS/Aura, first presented a detailed comparison between the altitude structures of all (eastward and westward travelling) QTDWs extracted from both the synoptic NOGAPS-ALPHA data and the asynoptic MLS/Aura ones to determine the aliasing. Hence that satellite data can be used quite successfully for studying planetary waves.

Sorry, I have never mentioned that the MLS/Aura data can be used for investigating the non-linear interaction between the tides and planetary waves; Aura is sun synchronized satellite and the tides cannot be determined at all.

- Thank you very much for these informations. We will include the Pancheva et al. 2018 paper citation and make a note on the sun synchronized Aura MLS satellite.

- P10, L14: Included the citation of Pancheva et al. 2018.

- P10, L16 (Conclusions): ...like tides or semi-diurnal oscillations, which cannot be resolved by Aura MLS because it is a sun synchronized satellite.

Please, see below only part of the papers based on ground-based measurements where the non-linear interaction between tides and quasi-2-day wave is studied. If in the title is mentioned planetary waves this means that besides the 2-day wave other PW waves are considered as well.

- This is a very nice overview of papers dealing with non-linear interactions between tides and quasi-2-day waves. We will take some of those to cite them in our manuscript.

- P2: "Moreover a high temporal resolution offers the possibility of investigating non-linear wave wave interactions between Q2DW and atmospheric waves with even shorter periods, like diurnal or semi-diurnal tides (Pancheva, 2001; Kumar et al., 2008; Guharay et al., 2015)."

Finally, I would like to clarify that the sum secondary wave generated by the non-linear coupling between the quasi-2-day wave and the diurnal tide should be 16-hour wave; the 18-h wave can be generated between the 3-day wave and the diurnal tide.

- Most of the time we wrote about a quasi 18-hour wave (or a wave close to 18 hours) where the 16-hour mode would be included. So far we have not looked at the 3-day wave, but it could be interesting for future investigations. By now no changes are made regarding this statement.

[revised manuscript text omitted]